# Case Report of Misleading Features of a Rare Sertoli Cell Testicular Tumor

**DOI:** 10.3390/medicina55050170

**Published:** 2019-05-20

**Authors:** Marius Anglickis, Rokas Stulpinas, Giedrė Anglickienė, Justinas Gabrilevičius, Arūnas Jaškevičius

**Affiliations:** 1Vilnius City Clinical Hospital, Department of Urology, 10207 Vilnius, Lithuania; justinas.gabrilevicius@gmail.com; 2National Center of Pathology, Affiliate of Vilnius University Hospital, 08406 Vilnius, Lithuania; rokas.stulpinas@gmail.com; 3National Cancer Institute, Department of Chemotherapy, 08406 Vilnius, Lithuania; giedre.anglickiene@gmail.com

**Keywords:** testicular tumor, Sertoli cell tumor

## Abstract

Testicular Sertoli cell tumors are extremely rare. Generally, they are benign neoplasms, which belong to a group called sex cord–stromal tumors. In this article, we present a case report of a Sertoli cell tumor, which was accidentally discovered during a urological consultation of a 42-year-old male. An ultrasound showed a 2.1 × 2.2 cm hypoechogenic, hypervascular tumor in the middle third of the left testicle. Serum tumor markers (α-fetoprotein, alkaline phosphatase, β-human chorionic gonadotropin, and lactic dehydrogenase) were all within the normal range. Rapid microscopic evaluation of fresh frozen sections during the operation was inconclusive, which led to a decision not to perform a radical orchiectomy immediately. On formalin-fixed paraffin-embedded (FFPE) sections, the tumor histology showed atypical patterns, and immunohistochemical analysis was performed in order to determine the type of neoplasm and differentiate it from other types of testicular tumors, so as to assign the further course of treatment. Radical inguinal orchiectomy was performed. The final pathology report showed a tumor with no predictive signs of aggressive behavior, which most closely resembled a Sertoli cell tumor.

## 1. Introduction

Sex cord–stromal neoplasms comprise less than 5% of testicular tumors. Data from the US National Cancer Database, published in 2016, showed that only 0.39% of patients (315/79, 120) were diagnosed with primary malignant Sertoli cell or Leydig cell tumors [1]. Only 65 (21%) of these patients had a malignant Sertoli cell tumor. One- and five-year overall survival rate for stage I Sertoli cell tumors was 93% (95% CI (confidence interval): 83–100) and 77% (95% CI: 62–95), respectively. Sertoli tumors are usually found by chance as painless scrotal masses without the presence of any other symptoms. On an ultrasound, a Sertoli cell tumor appears as a hypoechoic intratesticular lesion which is usually solitary [2]. These tumors are usually benign, but a few cases of malignancy have been observed. Regardless of that, the usual treatment is a radical inguinal orchiectomy; additional treatment is only needed if surgery is not radical or distant metastases are found. In this report, we present an unusual case of a rare testicular tumor exhibiting atypical features, which proved challenging to diagnose.

## 2. Case Report

The patient, a 42-year-old man, seeking a vasectomy operation, was consulted by a urologist. The patient did not have any symptoms specific to the urogenital system. The ultrasound scan showed a 2.1 × 2.2 cm hypoechogenic, hypervascular tumor in the middle third of a left testicle (Figure 1). Previous cryptorchidism was not reported. The patient had had a testicular trauma 3 months before. The family history was negative for any neoplasms. There were no physical signs (i.e., gynecomastia, etc.) of a hormone imbalance observed. Serum cancer markers (α-fetoprotein, alkaline phosphates, β-human chorionic gonadotropin, and lactic dehydrogenase) were all within the normal range. As diagnosis was not clear, it was decided to perform a rapid microscopic evaluation. Rapid microscopic evaluation of fresh frozen sections during the operation was inconclusive; hence, a radical orchiectomy was not performed immediately. On formalin-fixed paraffin-embedded (FFPE) sections, the tumor histology showed atypical patterns, and immunohistochemical analysis was performed in order to determine the type of neoplasm and differentiate it from other types of testicular tumors so as to assign the further course of treatment. A full-body CT (computed tomography) scan showed no evidence of metastatic disease; thus, a radical inguinal orchidectomy was performed. The gross examination found the tumor to be of similar color to the rest of the testicular tissue but of firmer texture. Histological analysis revealed that tumor had a biphasic structure (Figure 2) and was composed of a hypocellular collagenous stroma and solid nested serpentine trabecular structures (with small scant tubule formation and lumina containing homogeneous eosinophilic secretion (Figure 3)) from small to medium size cells with pale eosinophilic, finely vacuolated cytoplasm, and evenly centered round nuclei with a small peripheral nucleolus, finely dispersed chromatin, and unidentifiable mitotic activity. Usually, when an indolent epithelioid testicular tumor (most probably primary) is discovered in a middle-aged patient, the sex cord–stromal tumor group is the first one to turn to; therefore, an initial array of immunohistochemistry stains (based on WHO classification) was ordered. The tumor showed positive for Beta-Catenin (Figure 4) and CD99 (Figure 5); Ki67 (Figure 6) proliferative activity was very low ~1% (0.987% using Aperio “Nuclear v9” algorithm). As CD99 was the only typical positive “sex cord” marker, additional stains were ordered to clarify the case and exclude other malignancies (see Table 1).

In conclusion, the histologic pattern and the immunophenotype are not entirely typical but most closely resemble a Sertoli cell tumor. Permission was issued by Vilnius City Clinical Hospital of Medical Ethics Commission (Nr. V6-4, 2019-03-02). Informed consent was obtained from the participant.

## 3. Discussion

Testicular neoplasm by itself is a rare condition, accounting for only 1% of all neoplasms in men. Neoplastic Sertoli cells are very rare and account for less than 1% of testicular tumors. The tumor in patients with an enlarged testis is usually found accidentally on an ultrasound scan, as in our case. Most of the Sertoli cell tumors have a benign clinical course, but 10% to 22% of these tumors can change their behavior to aggressive. These malignant tumors are usually characterized by large (>5 cm) pleomorphic nuclei with nucleoli, increased mitotic activity (>5 per 10 HPF (high power fields)), areas of necrosis, and vascular invasion [4,5]. Sertoli cell tumors may occur in any age group, but they are the most common in adult males. The average age of a patient diagnosed with a Sertoli cell tumor is 45 years [6]. The molecular mechanisms involved in the etiopathogenesis of sex cord tumors are unclear. Recently, reports from immunohistochemical assay and mutational analysis of exon 3 of the *CTNNB1* gene by direct sequencing have shown a mutation in beta-catenin, a protein involved in the WNT signaling pathway. Sertoli cells of normal testis express beta-catenin and SOX-9 (a transcription factor), while the mutated sex cord tumors show nuclear immunopositivity for beta-catenin along with cyclin D1 [7]. Various forms of Sertoli cell tumor have been described in the literature. The main four among them are the classic Sertoli cell tumors, large-cell calcifying tumors with characteristic calcifications, sclerosing, and NOS (not otherwise specified) tumors.

In this case, a tumor did not exhibit any of the criteria for malignant behavior (size >5 cm, extratesticular spread, prominent cytological atypia, necrosis, high mitotic activity or lymphovascular invasion). However, neither the histologic pattern nor the immunophenotype s were entirely typical for a Sertoli cell tumor, and we were left with only a few options to choose from. Diagnosis of a “mixed sex cord–stromal tumor” should be avoided as only the epithelioid component with no stromal counterpart is clearly identifiable. The tubules seen in this tumor remind of a “sex cord tumor with annular tubules (SCTAT)”, a distinctive neoplasm with indifferent cells of sex cord derivation in a characteristic arrangement of ring-like tubules. SCTAT has been placed under an “unclassified sex cord-stromal” category in the World Health Organization (WHO) Classification [8] of ovarian tumors (but is not listed among testicular tumors at all). A term of “unclassified sex cord tumor” could be more appropriate, as WHO lists them under such a description: “The unclassified tumors have variable components of epithelial cells of sex cord type with patterns that do not lend themselves to subclassification as Sertoli or granulosa cell tumor. There is often conspicuous fibrous stroma. Their immunohistochemical profile parallels that of Sertoli or granulosa cell tumors, but unclassified tumors may be less positive for typical sex cord-stromal markers.” Thus, it seems that a specific formulation of diagnosis remains a matter of personal decision, but a term of “unclassified sex cord tumor with predominant features of a Sertoli cell tumor” is a fine consensus in this case. At this moment, the prognosis of the patient is good due to lack of typical histological signs of malignancy, radical removal of the affected testis, non-elevated tumor markers, and no visible radiological signs of metastatic disease. Nevertheless, further regular follow-up is necessary for the detection and treatment of possible future metastasis. Understanding the types of Sertoli cell tumors helps pathologists and urologists to choose the correct therapeutic approach, as in our case, in which a radical inguinal orchiectomy was preferred. The immunohistological characterization is not always easy, and there might be repeated inconclusive cases. Testis-sparing surgery is probably the most appropriate therapy, but in the present case, this was not done because of uncertainties regarding the preliminary histology report. The present case report has the general aim to increase the knowledge about these rare tumors in order to let more patients benefit from conservative surgery. Nevertheless, recommendations for appropriate follow-up of Sertoli cell tumors cannot be given due to their rarity and the lack of follow-up data for most reported cases [9].

## 4. Conclusions

Due to their rarity and a comparatively small number of cases reported, Sertoli cell testicular tumors remain a relative mystery and a diagnostic challenge in modern medicine to this day. Our case further shows how an already rare tumor can present itself in a way which can lead to difficulties determining the final diagnosis and may affect subsequent treatment. Further research of these tumors should be encouraged in order to optimize the diagnosis, treatment, and follow-up of patients diagnosed with this illness.

## Figures and Tables

**Figure 1 medicina-55-00170-f001:**
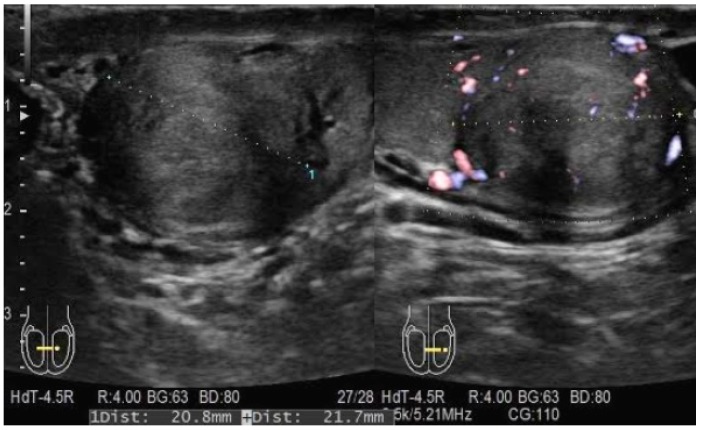
Hypoechogenic, hypervascular tumor in the middle third of the left testicle.

**Figure 2 medicina-55-00170-f002:**
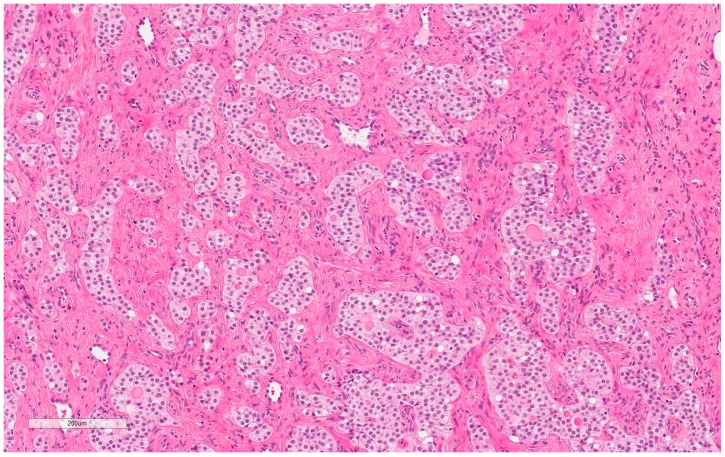
Hematoxylin–Eosin (100X): Nests and trabecular structures.

**Figure 3 medicina-55-00170-f003:**
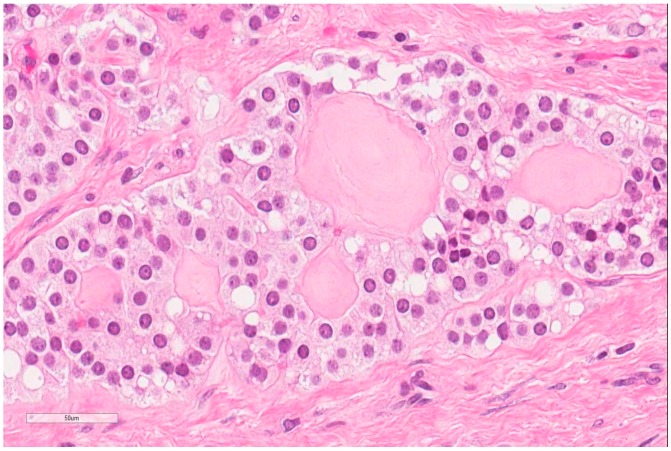
Hematoxylin–Eosin (400X): Nests and glandular structures composed of bland cells.

**Figure 4 medicina-55-00170-f004:**
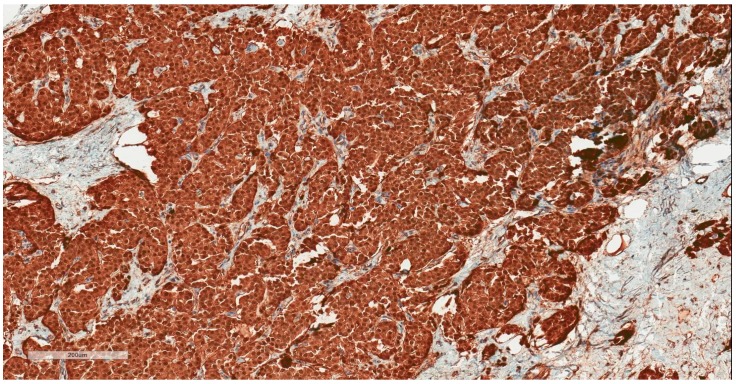
Beta–Catenin: Strong nuclear and cytoplasmic staining.

**Figure 5 medicina-55-00170-f005:**
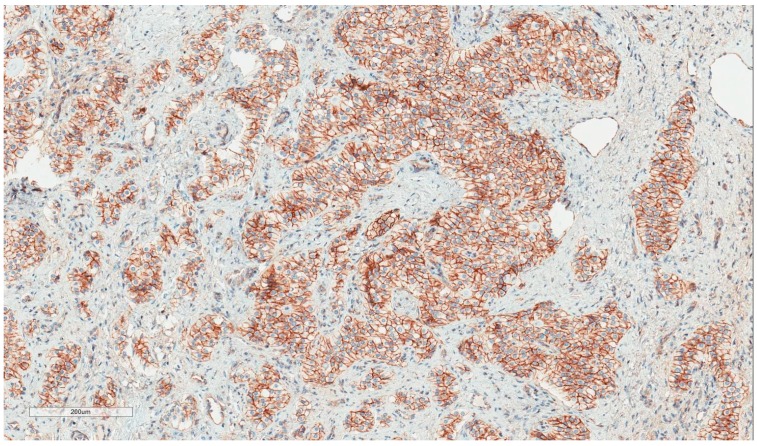
CD99: A field of strongest positivity.

**Figure 6 medicina-55-00170-f006:**
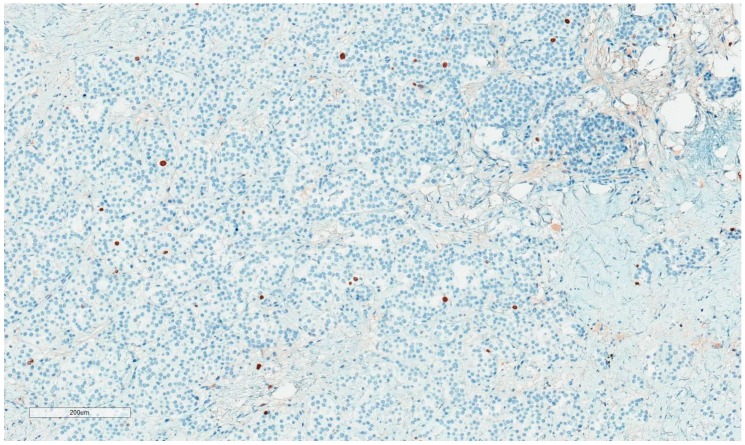
Ki67: Low proliferative activity.

**Table 1 medicina-55-00170-t001:** Immunohistochemical profile of the tumor.

Immunohistochemical Marker	Our Case	Typical Sertoli Cell Tumor [3]	Seminoma [3]
Vimentin	+++	positive	
Beta-Catenin	+++	positive (60–70% tumors)	
SF1	++	positive (“typically”)	
CD99	++	positive (“typically”)	
CD56	+		
S100	+	positive	
EMA	+/-	variable	negative
Synaptophysin	+/-	often positive	
Chromogranin A	-	often positive	
PanCytokeratin	-	often positive	variable (20–36% tumors)
Inhibin	-	positive (50% tumors)	
Calretinin	-	positive (“typically”)	
MelanA (A103)	-	positive (“typically”)	
PLAP	-		positive (90–100% tumors)
SALL4	-		positive (100% tumors)
OCT4	-		positive (100% tumors)

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
