# Peer review of "Case Report of Misleading Features of a Rare Sertoli Cell Testicular Tumor"

_medicina, 2019, doi:10.3390/medicina55050170_

Round 1

Reviewer 1 Report

The authors report the case of a benign testicular Sertoli cell tumour that was diagnosed by testis sparing excision. Full orchiectomy was subsequently performed because of uncertainties regarding the biological dignity of the neoplasm. The benign nature was only postoperatively established by extended immunohistological examination.

The authors document the immunohistological presentation of this tumour with positvity of beta catenin and CD 99 and a very low proliferative activity with Ki67.

The authors finally discuss the problems of clinical decision-making conferred by the histopathological difficulties of defining the biological dignity of the neoplasm.

The report is over-all interesting and worth being published, however, the key message of the paper needs to be elaborated more precisely. The key message is - at least in reviewer´s mind -  that this case would have been well served with testis sparing surgery alone (not full orchiectomy). Only due to the problems arising from the histopathologic characterisation of the neoplasm the second operation was performed. In the introduction the authors should point out that Sertoli cell tumours are excessively rare and that most of them are of benign nature. A good reference to support this connotation would be Featherstone JM (J. Urol 2009). Perhaps, a sentence regarding the typical appearance on ultrasound could be helpful (plus one reference).

Regarding the report itself, the reader would like to see a sonographic image of this patient. Also, the reader  is a little bit confused about the large number of immunohistochemical examinations that were either positive or negative. At this place a table would greatly increase the reader´s apprehension.

In the discussion the authors should (very briefly) summarize the world-wide knowledge about Sertoli cell tumours and compare their own case with that documented knowledge.  First summarize the data about clinical presentation and compare it with the presentation of the case. Then report ultrasound findings, compare with own findings. Then go to immunohistopathological findings, compare with the own findings. The conclusion should be that immunohistological characterisation is not always easy and there might always be inconclusive cases. Testis sparing surgery is probably the most appropriate therapy, but in the present case this was not done because of uncertainties regarding the preliminary histology report. The present case report has the general aim to increase the knowledge about these rare tumours in order to let more patients benefit from conservative surgery.

Minor points of critique

Introduction, line 30: say “US National Cancer Data Base” (otherwise one could think that the authors report from Baltic or Scandinavian registries)

Line 34: CI  - explain abbreviation “CI”

Line 34 “p=0.015)” This p-value should be deleted, it does not support valuable information given in the text.

Case report, line 56: “The sample was sent to a pathology center”. Delete this sentence because it is clear to any engaged reader that the surgical specimen needs to be thoroughly examined histologically.

Provide a table with all of the staining results

Discussion, line 73: This statement is not correct. Testicular neoplasms do not only comprise of germ cell tumours and sex cord gonadal stromal tumours. In addition, there are tumours of haematopoetic origin (e.g. lymphomas), tumours of the connective tissue (sarcomas and benign counterparts) as well as rete testis tumours and primitive neuroendocrine tumours (PNET, e.g. carcinoid). A good review to refer to at this place would be Mooney KL  et (Surg Pathol Clin 2018) and Idrees MT et al. The WHO classification …(Histopathology 2017).

Line 88 explain “NOS”

Line 126: the legend to figure 5 is obviously wrong. It must be Ki67: low proliferative activity (as correctly done in line 153)

References, line 178: The EAU guidelines published by Albers et al are best cited by: Albers et al. Eur. Urol. 68: 1054-68, 2015

Author Response

Thank you for your comments. We corrected all mistakes. 

Reviewer 2 Report

In this paper, the authors describe a case of Sertoli cell testicular tumor with an atypical histology. I is a short case report, written clearly, and discussing briefly other similar histopathologic entities. This type of tumor is still rather rare and any new piece of information may be welcome.

The English would benefit from revision and corrections.

Please, correct following:

1.       use “testicular tumor” or “Sertoli cell tumor” instead of “testicular cancer” or “Sertoli cell cancer”  - cancer is a malignancy arising from epithelial cells, which is not the case here

2.       similarly, use “(serum) tumor markers” instead of “(blood) cancer markers”

3.       in the row 33 omit “malignant”  - I assume the percentages apply for all Sertoli cell tumors, not only for those with a clear malignant potential (?)

4.       in the row 81 use rather “etiopathogenesis” or something like that instead of “histogenesis”

5.       in the Discussion (row 110-111),  it implies that an organ-sparing approach has been preferred and used in this case – which is not true, radical inguinal orchidectomy was performed (as reads in the row 54)

6.       in the Discussion (row 111-113), it cannot be stated that an organ-sparing approach is recommended for all small intraparenchymal testicular lesions until final histology – this may apply only for small lesions of unclear character;

if the lesion is considered to be a testicular germ cell tumor, then the standard (and only evidence-based) approach is a radical orchidectomy (testis-sparing surgery having very limited data and remaining rather an experimental approach!!!)

Author Response

Thank you for your comments. We correctedall mistakes. 

Reviewer 3 Report

The authors present a case report of a Sertoli cell tumor, which was accidentally discovered in a 42-year old man consulting for a vasectomy operation. These tumors are extremely rare (less than 1% of testicular tumors). Most of them are benign, but 10-22% can become aggressive. Their diagnosis remains challenging. In the present report, no histological sign of malignancy was found. Blood cancer markers (α-fetoprotein, alkaline phosphatase, β-human chorionic gonadotropin, lactic dehydrogenase) were within the normal range. Histological analyses performed on FFPE sections showed atypical patterns. Immunohistochemical analyses showed that the tumor was positive for CD99 (the only typical positive “sex-cord” marker), for β-catenin and vimentin. The authors conclude that the histological pattern and immunophenotype were not entirely typical, but closely resembled a Sertoli cell tumor. Although very descriptive, this case report presents an unusual case of a rare testicular tumor, that may help clinicians to better diagnose these rare conditions. The report is clear and well written. However, modifications are required to improve the manuscript:

1. The authors should explain briefly how they selected the different markers analyzed. Which cell types express these markers?

2. Figures 1 and 2 show HE staining of the tumor at different magnifications. They should therefore be included within the same figure.

3. Figure 5 shows Ki67 immunostaining and not β-catenin immunostaining. Please modify the legend.

4. The appendix is an optional section that can contain supplemental data. However, the figures presented in Appendix B are the same as those presented in the text.

5. Several typing errors need to be corrected:

Lines 19 and 47: Please replace “markes” by “markers”

Lines 19 and 47: Please replace “phosphates” by “phosphatase”

Line 61: Please replace “innitial” by “initial”

Line 77: Please replace “teir” by “their”

Line 105: Please replace “consenssus” by “consensus”

6. Please define the following terms:

Line 53: Please define CT

Line 79: Please define HPF

Line 88: Please define NOS

Author Response

Thank you for your comments. We correctedall mistakes. We would suggest leaving Fig.1 and Fig.2 separated, because although they both use the same hematoxylin-eosin stain, they have different purposes: first photo shows a panoramic picture of biphasic ("red vs. pale") tumor pattern and the second close up photo shows bland, monotonous cytologic details and scant tubule formation.

Round 2

Reviewer 1 Report

The manuscript has been improved accordimng to the comments of the reviewer although not all of the recommendations hacve been adopted.

Nonetheless, the manuscript is now ready for acceptance.